# Indoor Particulate Matter in Urban Households: Sources, Pathways, Characteristics, Health Effects, and Exposure Mitigation

**DOI:** 10.3390/ijerph182111055

**Published:** 2021-10-21

**Authors:** Ling Zhang, Changjin Ou, Dhammika Magana-Arachchi, Meththika Vithanage, Kanth Swaroop Vanka, Thava Palanisami, Kanaji Masakorala, Hasintha Wijesekara, Yubo Yan, Nanthi Bolan, M. B. Kirkham

**Affiliations:** 1Nantong Key Laboratory of Intelligent and New Energy Materials, Nantong University, Nantong 226019, China; aling2017@jsfpc.edu.cn; 2School of Health, Jiangsu Food & Pharmaceutical Science College, Huai’an 223003, China; 3Molecular Microbiology and Human Diseases Project, National Institute of Fundamental Studies, Hantana Road, Kandy 20000, Sri Lanka; dhammika.ma@nifs.ac.lk (D.M.-A.); meththika@sjp.ac.lk (M.V.); 4Ecosphere Resilience Research Center, Faculty of Applied Sciences, University of Sri Jayewardenepura, Nugegoda 10250, Sri Lanka; 5Priority Research Centre for Healthy Lungs, Faculty of Health and Medicine, School of Biomedical Sciences and Pharmacy, The University of Newcastle, Callaghan, NSW 2308, Australia; kanthswaroop.vanka@uon.edu.au; 6Global Innovative Centre for Advanced Nanomaterials (GICAN), Faculty of Engineering and Built Environment, The University of Newcastle, Callaghan, NSW 2308, Australia; thava.palanisami@newcastle.edu.au; 7Department of Botany, Faculty of Science, University of Ruhuna, Matara 80000, Sri Lanka; mas@bot.ruh.ac.lk; 8Department of Natural Resources, Faculty of Applied Sciences, Sabaragamuwa University of Sri Lanka, Belihuloya 70140, Sri Lanka; wijesekara@appsc.sab.ac.lk; 9Jiangsu Engineering Laboratory for Environment Functional Materials, Huaiyin Normal University, Huai’an 223300, China; 10School of Agriculture and Environment, Institute of Agriculture, The University of Western Australia, Perth, WA 6001, Australia; Nanthi.Bolan@uwa.edu.au; 11Department of Agronomy, Kansas State University, Manhattan, KS 66506, USA; mbk@ksu.edu

**Keywords:** indoor particulate matter, environmental health, exposure, mitigation

## Abstract

Particulate matter (PM) is a complex mixture of solid particles and liquid droplets suspended in the air with varying size, shape, and chemical composition which intensifies significant concern due to severe health effects. Based on the well-established human health effects of outdoor PM, health-based standards for outdoor air have been promoted (e.g., the National Ambient Air Quality Standards formulated by the U.S.). Due to the exchange of indoor and outdoor air, the chemical composition of indoor particulate matter is related to the sources and components of outdoor PM. However, PM in the indoor environment has the potential to exceed outdoor PM levels. Indoor PM includes particles of outdoor origin that drift indoors and particles that originate from indoor activities, which include cooking, fireplaces, smoking, fuel combustion for heating, human activities, and burning incense. Indoor PM can be enriched with inorganic and organic contaminants, including toxic heavy metals and carcinogenic volatile organic compounds. As a potential health hazard, indoor exposure to PM has received increased attention in recent years because people spend most of their time indoors. In addition, as the quantity, quality, and scope of the research have expanded, it is necessary to conduct a systematic review of indoor PM. This review discusses the sources, pathways, characteristics, health effects, and exposure mitigation of indoor PM. Practical solutions and steps to reduce exposure to indoor PM are also discussed.

## 1. Introduction

Air pollution results from introducing various contaminants into the atmosphere that are likely to be detrimental to humans, other living organisms, and the natural environment [1]. Among a wide range of air pollutants, particulate matter (PM) is of particular concern because of its association with cardiopulmonary health disorders [2,3]. PM is a complex mixture of solid particles and liquid droplets made up of metals, organic compounds, sulfate, nitrate, ammonium, and other ions (Figure 1) [2,4]. In different indoor environments, the composition is generally the same, but due to different indoor environment types, the difference may be small, and the proportion may be significantly different [5,6,7,8,9]. Early studies have shown that PM’s composition is associated with some respiratory diseases, including asthma, chronic bronchitis, and acute bronchitis [10,11,12].

In addition to physical and chemical composition, the size of PM is an essential factor related to its effect on health. Briefly, PM is broadly categorized by its “aerodynamic equivalent diameter” (AED). As shown in Figure 2, particles with diameters between 2.5 and 10 μm (PM_2.5–10_) are defined as “coarse”; less than 2.5 μm as “fine”; and less than 0.1 μm as “ultrafine” [13]. The main threat to health from PM is inhalable particulate matter, which can penetrate the chest area of the respiratory system and cause adverse health effects (Figure 3) [14]. For example, because PM_2.5_ is light, it has a higher incidence rate and deposition rate in the lungs than other particles, resulting in it staying longer in the respiratory tract [15,16]. Pope et al. found that for every 10 μg/m^3^ increase in PM_2.5_ levels, the risk of death increases by 8% to 18% [16]. According to data from the World Health Organization, about seven million people die from exposure to PM_2.5_ in polluted air every year [17]. Moreover, ultrafine particles cause a more significant inflammatory response than fine particles per given mass [18], and the ultrafine-particle-toxicity effect can be enhanced by a gaseous co-pollutant such as ozone [14,19].

With technological advancements and lifestyle changes, more human activities, including cooking, cleaning, and indoor sports activities, are carried out indoors, all of which are likely to increase indoor PM [20]. Exposure studies show that indoor PM contributes substantially to personal exposure, and the indoor PM concentration levels may exceed those outdoors [1]. Specifically, epidemiologic studies have presented evidence that indoor PM plays a significant role in human health, such as lung malfunctioning, cardiovascular disease, respiratory symptoms, asthma, and premature births [21,22,23,24]. In addition, PM may alter the immune response by promoting immunoglobulin E, causing an inflammatory response. Nowadays, the indoor, suspended PM concentration has been identified as one criterion for evaluating indoor environmental quality [25]. For example, in 2012, Canada established the Residential Indoor Air Quality Guidelines, which state that PM_2.5_ needs to be monitored, with a limit of 100 µg/m^3^ as a 1 h average (Short-Term Exposure) and 40 µg/m^3^ as an 8 h average (Long-Term Exposure). The National Ambient Air Quality Standard (NAAQS), set by USEPA, stipulates 35 µg/m^3^ and 15 µg/m^3^ for 24 h and annual periods, respectively, for exposure to PM_2.5_. Furthermore, the World Health Organization recommended to apply to indoor spaces the same PM guidelines as for ambient air, presented on the 2005 global update, which are 25 and 50 μg m^−3^ for PM_2.5_ and PM_10_, respectively (over 24 h) [26].

Indoor PM sources include indoor origins and outdoor infiltration. Primary indoor sources of PM result from specific activities (cooking, sweeping, dusting, candle or incense burning, using laser-printing devices, fuel combustion for heating, and smoking tobacco), the design of the house, and secondary organic aerosols [27,28,29,30]. Due to air exchange, the indoor PM also originates from outdoor sources, including natural ones (forest fires, soil dust, and sea salt) and anthropogenic ones (transport, oil combustion, and coal burning in power plants) [2,31]. The automatic monitoring methods of PM_2.5_ ambient air quality are the β-ray method and micro-oscillatory balance method. The effectiveness of urban ambient air quality assessment can be effectively improved by selecting suitable methods in suitable areas.

To ensure the health and comfort of indoor environments, strategies should be taken to reduce indoor PM concentrations. The development of technology has allowed the new trend of operating indoor air purifiers to control indoor PM concentrations. Systems based on the principle of electrostatic precipitation and air purifiers using filtration technologies are two of the most common indoor air-purification technologies [32,33,34]. Natural ventilation with open windows is a common and economical approach to diluting indoor PM [35].

Numerous studies have described sources and health risks of indoor PM, but their conclusions have been focused on a single topic, e.g., the sources and its effects on susceptible subgroups [36,37,38]. As the quantity, quality, and scope of the research have expanded, this review is necessary to discuss new information, e.g., the characteristics, distribution, and pathways. This paper conducts a systematic review of sources, pathways, characteristics, and health effects of indoor PM. In addition, practical steps to reduce exposure to indoor PM are also discussed.

## 2. Sources and Distribution of Indoor PM

### 2.1. Major Sources of Indoor PM

Indoor particulate pollution is grouped into primary and secondary PM, based on the origin of the PM. Primary indoor pollutants are directly generated from indoor domestic activities such as cooking, biomass heating, tobacco smoking, washing, cleaning, and other indoor activities. Secondary PM includes pollutants infiltrated from the outdoor environment and particles generated due to chemical reactions between indoor precursors and outdoor sources [39,40]. It is well known that ambient (outdoor) PM is a significant contributor and determining factor of indoor PM levels. Other factors, including indoor-type homes, offices, and commercial spaces; ventilation arrangements (naturally provided by windows or mechanical ventilation); occupancy rate and time; endotoxin levels; and geographical location, play a critical role in defining the chemical composition and disease burden of indoor PM [41,42]. According to the study of Sumpter and Chandramohan, the majority of lower-income groups from developing and low income countries rely on solid fuels for cooking and heating [43]. Activities such as cooking and heating using biofuels (coal and wood) can generate significant indoor PM concentrations, especially PM_2.5_ and ultrafine particulate matter [44]. Additionally, indoor exposure to asbestos fibers in old houses has been highlighted as a significant concern [45,46]. Due to its high tensile strength and versatility, asbestos, including crocidolite, was once widely used in construction (roofing, floors, and walls) and manufacturing household items such as fireproof curtains. Over time, weathering releases microscopic fibers, which are highly fibrogenic to the human lungs upon inhalation [45]. Furthermore, human habits, such as frequent windows and other dust-generating indoor activities, result in crucial indoor PM sources. For example, human walking is an important factor causing the resuspension of indoor PM. During the movement of humans, soles were exposed to the air. Branis et al. found that indoor human activities in the classroom could also lead to resuspension of large particulate matter, especially for PM_10_ [47].

Indoor cooking has been a well-investigated source of PM over the past decades. Studies have shown that cooking activities enable the emission of millions of particles (~10^6^ particles/cm^3^) through oil, wood, and food combustion, and most of them are ultra-fine particulates [48,49]. In addition, cooking can lead to indoor PM emissions from cooking areas in homes, restaurants, and other building types (offices, schools, etc.) because high-temperature cooking can lead to water vapor and other solid and liquid particle emissions.

High emissions of indoor fine particulates (aerodynamic diameter <100 nm) occur during frying and boiling [50]. Zhang et al. showed that average concentrations of ultrafine, fine, and black carbon particles emitted during boiling and frying ranged from 1.34 × 10^4^ to 6.04 × 10^5^ particles/cm^3^, 10.0 to 230.9 μg/m^3^, and 0.1 to 0.8 μg/m^3^, respectively [51]. Chinese-style cooking has been identified as one of the major sources of indoor PM. It contributes up to 33% of indoor PM levels (PM_0.5-5_) [52]. The PM exposure level of indoor populations due to cooking and heating is related to fuel type, stove type, and population type [53,54,55]. Cooking behavior, such as using different types of aerosols, fuels, and exhaust fans or stove hoods, is also closely linked to the level of indoor PM pollution. A study showed that, when no stove hood was used, indoor PM_2.5_ was at a sufficient concentration to affect potentially the health of children, even in non-cooking times [56].

Tobacco smoke from cigarettes, water pipes, and e-cigarettes is also a well-investigated source of indoor PM, which threatens the well-being of both smokers and other occupants. Drago et al. showed that concentrations of PM_2.5_ and several toxic trace elements were higher in smoker dwellings than non-smoker dwellings [57]. Braun et al. studied the effects of tobacco strength, measured by quantifying the amount of tar, nicotine, carbon monoxide, and different additives on the amount of PM [58]. This study included five cigarette types with different tobacco strengths, with or without additives, and a reference cigarette. Studies have found that incense is an important source of polycyclic aromatic hydrocarbons (PAHs), carbon monoxide, benzene, isoprene, PM_2.5_, and PM_10_ [5]. Lee and Wang observed that PM_2.5_ emission rates of different incense types varied considerably [59]. PM levels were directly proportional to concentrations of residuals in the cigarettes. Compared to tobacco smoke, incense smoke contains higher concentrations of PM. Lin et al. showed that incense burning generates PM >45 mg/g compared to 10 mg/g from cigarette burning [60]. Kumar et al. investigated PM levels generated from indoor incense-smoking activities, such as pre-burning, burning, and post-burning phases [61]. Incenses that they studied included sandalwood and floral sticks, dhoops (dhoops are an extruded incense, lacking a core bamboo stick), and mosquito coils (a mosquito coil is a mosquito-repelling incense). They showed that the mean concentrations of PM during the burning phase were highest. Concentrations ranged from 1300 to 1880 μg/m^3^ for dhoops and from 214 to 259 μg/m^3^ for mosquito coils. The burning of floral incense had a higher PM concentration (700 to 854 μg/m^3^) compared to the burning of sandalwood incense (99 to 114 μg/m^3^). With the increasing popularity of IQOS and e-cigarette devices among adolescents and adults, more and more studies have been conducted to explore the relationship between PM_2.5_ and e-cigarette aerosol. Studies have shown that indoor PM_2.5_ concentrations can rise to 197–818μg/m^3^ during vaping [62]. The level of PM_2.5_ is comparable to or even higher than those found in conventional cigarettes. IQOS smoking had little effect on indoor fine particulate matter (>300 nm) concentration or PM_2.5_ concentration. However, the concentration of ultrafine particles (25–300 nm) can be significantly increased [63]. Overall, PM emission during the burning of dhoops was higher than PM emission during the burning of sandalwood or floral incense and mosquito coils. This study also showed a higher PM emission during the post-burning phase of dhoops than sandalwood and floral incenses and mosquito coils, which indicated potential exposure even after the cessation of the burning phase of dhoops.

Besides indoor in-situ sources, outdoor PM can be transported into indoor environments through air movement. Particles in the outdoor environment enters the room through air flow, that is, through a combination of osmosis, natural (NV) and mechanical ventilation (MV). PM originates from a range of outdoor natural sources, including forest fires, soil dust, sea salt, the presence of pets and farm animals (i.e., animal debris), pollen, spores, plant debris, and bacteria. Anthropogenic outdoor PM sources, such as evaporative gasoline emissions from transport, oil combustion, and coal burning in power plants, contribute to indoor PM concentrations [2,31].

### 2.2. Distribution Characteristics of Indoor PM

Distribution characteristics of PM include size, mass, mass, or particle concentration, and chemical or biological composition. The mass of PM is a major characteristic that affects its distribution. Microscopic particles with low density can remain airborne for extended periods and move freely from source to surrounding areas, reducing indoor and outdoor air quality [64]. Indoor PM burden may range from 15 to 259 μg/m^3^ and from 3 to 202 μg/m^3^ for PM_10_ and PM_2.5_, respectively [65]. Factors such as geographical location, air exchange efficiency, penetration and deposition rate, occupancy rate, and the particles’ presence dictate indoor PM levels. Not all indoors are affected equally. For example, residents of a building living closer to ground level experience higher PM levels compared to occupants residing at higher levels [66].

Similarly, individuals located in cities with frequent dust storms have elevated indoor PM levels and are more susceptible to PM-associated diseases compared to residents of other cities [67,68]. The effect of human exposure to PM is especially significant in urban environments, where higher population density leads to higher pollutant generation and higher human exposure [69]. Densely populated urban centers curb black carbon emissions from fossil-fuel transport, household stoves, and space heating. The negative effects of this growing pollution have been enhanced by the continuous movement of people from rural to urban areas. Ventilation of the confined spaces plays a critical role in regulating indoor PM levels. Spaces equipped with mechanical ventilation or poor air conditioning systems have high infiltrated PM_2.5_ levels from outdoors.

Chithra and Shiva Nagendra measured the temporal characteristics of PM concentrations inside a room in a naturally ventilated school building located near a roadway in Chennai, India [70]. They found that, during working hours, the number concentrations of PM inside the room were 2.4 × 10^5^, 2.2 × 10^3^, and 8.1 × 10^2^ particles/dm^3^ in size ranges of PM_0.3–1_, PM_1–3_, and PM_3–10_ µm, respectively. Putaud et al. showed a difference in the chemical profiles of coarse and fine particulate fractions of aerosol PM [71]. The coarse PM was made up of mineral composites (e.g., crustal species, fly ash, and minor elements), sea salt, and black carbon. Fine particulates contained ammonium sulfate, ammonium nitrate, and organic compounds. The formation of secondary aerosols is a major factor that affects PM pollution. Both secondary organic and inorganic aerosols can be dominant components of PM. Wang et al. found secondary inorganic species, including sulfate, nitrate, and ammonium, in PM air samples collected from four regions in China [72]. They also showed the production of secondary organic aerosol species in three regions and suggested that aqueous-phase processing and photochemical reactions were occurring and they were possible causes for the production of the organic aerosols.

In addition to mass and size, biological composition is another important characteristic of indoor PM distribution. Indoor PM biological compositions, also known as bioaerosols, are solid or liquid particles carrying living organisms from biological sources, with sizes ranging from 0.1 mm to 100 mm in diameter [73]. Generally, their particle size distribution varies from the nucleation mode (<30 nm in vacuum cleaning condition) to the accumulation mode (~100 nm, indoor combustion aerosols from smoking, cooking, or incense burning), and to the fine and coarse modes (>1 µm, resuspension aerosols) [74,75]. They include biological allergens (e.g., animal dander and cat saliva, house dust, cockroaches, mites, and pollen) and microorganisms (viruses, fungi, and bacteria) [76,77,78]. Biological allergens, known as antigens, originate from a number of insects, animals, mites, plants, or fungi, and will induce an allergic state in reacting with specific immunoglobulin E antibodies. Indoor sources of allergens mainly include furred pets (dog and cat dander), house dust mites, molds, plants, cockroaches, and rodents [79], and there are outdoor sources as well [80]. Humans and animals are one of the dominant sources of bacteria in indoor environments, while fungi mostly originate from the outdoor environment [81,82,83].

### 2.3. Factors Influencing the Distribution of Indoor PM

Both indoor and outdoor environmental conditions influence indoor PM levels. Indoor factors, such as temperature, humidity, and air exchange rate and efficiency, are some of the important ones that dictate the indoor PM levels. Outdoor factors, such as weather, wind velocity, temperature, humidity, and solar radiation, influence indoor PM levels and secondary aerosol formation. For example, changes in temperature affect PM by influencing the change of chemical reaction rates and atmospheric mixing heights that affect the vertical dispersion of pollutants and modifying local wind and flow patterns that control the transportation of pollutants [84]. Meanwhile, differences in temperature indoors and outdoors also influence natural ventilation through the movement of air, and thus affecting indoor PM concentration [85]. With the increase of relative humidity, the resuspension rate of fine particulate matters decreased [86]. Several studies have demonstrated that air exchange rate has a significant effect on indoor PM concentrations under stable outdoor PM concentrations. In general, the higher air exchange rate was, the lower the indoor PM concentration was [87]. Seasonality is a major factor influencing the distribution of indoor PM. Indoor PM_2.5_ concentration is related to season and building type. The PM concentration in the heating season was significantly higher than that in the non-heating season [88,89]. Epidemiological data from urban and rural areas in developing and developed countries show a steep rise in airborne PM during autumn and winter compared to spring and summer. Although advances in technology have provided safe, conventional (electrical) cooking and heating methods in developed countries, the majority of the population from urban and rural areas in developing countries still relies on traditional heating (burning wood and coal) to keep homes warm. Huang et al. studied three urban areas of China (Beijing-Tianjin-Hebei) between 2013 and 2017 and showed that the average concentration of outdoor PM_2.5_ in the springtime (July) was 38.76 mg/m^3^; by winter (January), levels had risen to 133.10 mg/m^3^, which was 5.3 times above WHO air quality standards [90,91]. The use of conventional heating methods, such as gas and electricity, reduced indoor levels of PM_2.5_ in heating and non-heating seasons by 43% and 70%, respectively [54].

Meteorological factors during different seasons, such as wind velocity, precipitation, air and soil temperatures, and atmospheric and soil humidity, have noticeable relationships with the PM distribution. Deng et al. measured PM at heights of 121 and 454 m on the Canton Tower in China and showed that the vertical distributions of PM decreased with height [92]. Chen et al. studied the effects of SO_2_ and NH_3_ on secondary aerosol formation from unburned gasoline vapor and found that an increase in SO_2_ and NH_3_ concentrations (from 0 to 151 ppb and from 0 to 200 ppb, respectively) promoted the formation of secondary aerosols by a factor of 1.6 to 2.6 and 2.0 to 2.5, respectively [93]. They also reported that new particle formation and particle size growth were enhanced in SO_2_ and NH_3_. Increased solar radiation and intensity promote photochemical reactions that lead to the formation of secondary aerosols, thereby accelerating PM pollution and its distribution in the environment [72].

### 2.4. Unique Characteristics and Spatial-Temporal Distribution of Indoor PM

PM possesses unique characteristics, such as a small aerodynamic diameter, a large surface to volume ratio, a complex chemical profile, and a high toxicity index, which depend upon the origin and source of the PM. The large surface to volume ratio provides a natural platform for reactive chemicals and ionic species to undergo oxidative and reduction reactions (RedOx). The large surface area of the particles can harbor surrounding environmental pollutants and acts as a “carrier” for heavy metals (e.g., Cr, Pb, Hg, Cd, and As), organic pollutants such as polyaromatic hydrocarbons (PAH), heterocyclic amines, and inorganic minerals (e.g., Si, Al, Fe, Mn, Ca, Cl, and Zn) [89,94,95]. The microscopic size and aerodynamic diameter allow the particles or particle-bound contaminants to remain airborne and drift along with the winds to reach extreme distances. The chemical composition of indoor PM is complex because it comprises particles from indoor and outdoor origins. Indoor activities, such as cooking using solid fuels including coal, wood, dung, and kerosene, can liberate black-carbon soot (elemental carbon) and organic carbon (bound carbon). Other major factors affecting both indoor and personal exposure to PM include the penetration factor, air exchange rate, and particle deposition and sedimentation, as well as human behavior [96]. Hung et al. studied indoor PM_2.5_ and PM_10_ in office spaces and showed an increase of 0.211 μg/m^3^ and 0.226 μg/m^3^ per 1 μg/m^3^, respectively, over outdoor levels [97]. Climatic and environmental factors affect both indoor and outdoor PM levels and composition.

Annual average levels of PM_10_, PM_2.5_, and PM_1_ at observational heights of 121 m and 454 m above ground were 44.1, 38.2, and 34.9 μg/m^3^ and 35.7, 30.4, and 27.5 μg/m^3^, respectively [98]. Spatio-temporal studies conducted on PM_10_, PM_2.5_, and PM_1_ levels in Seoul (Korea) metropolitan subways showed a vertical distribution (going upwards from the ground) in the sequence PM_10_ > PM_2.5_ > PM_1_, and levels of fine particulates were uniform compared to coarse particles [99]. Moreover, the distribution was constant in summers compared to other seasons. Air quality monitoring studies conducted in confined and crowded spaces such as underground subways and metro stations often record a several-fold increase (4- to 6-fold) in indoor PM levels compared to ambient levels [99,100,101]. Human activities, such as primary or second-hand smoking, add airborne PM and nicotine to indoor microenvironments [9].

## 3. Pathways of Exposure to Indoor PM

As shown in Figure 4, potential sources for indoor PM include bioaerosols (plant, animal, bacteria, fungi, and viruses), combustion of fuel used for cooking and heating, and home or personal care products [102,103,104]. The diameter of these particles is in a range of 0.001–2000 µm. Even though some consider that the amount of PM is more significant indoors than outdoors, the concentrations vary with location (e.g., urban or rural or proximity to roadsides) and with socio-economic factors of the population [104,105,106,107]. Smoking, cooking using gas and wood stoves, and cleaning are the major sources for elevated indoor levels of PM_10_ and PM_2.5_ [108]. This PM can enter the human body through inhalation, dermal absorption, or ingestion [106,109,110].

### 3.1. Respiratory Absorption

Respiratory absorption is the most common route of exposure. Both in an outdoor or indoor environment, PM in the air can get into the body via breathing through the nose, which is able to filter large particles, or the mouth, which is unable to perform that particular task [111,112,113]. Ample evidence demonstrates that the “personal cloud,” which is resuspended house dust, is one of the major pathways of exposure for inhalation [113]. Passing through the pharynx and larynx, air-containing particles enter the trachea, which is connected to the left and right primary bronchi. Primary bronchi are further subdivided into bronchioles leading into alveolar ducts and sacs in the lung. It is estimated that particles up to 100 μm in aerodynamic diameter (Dae) present in the inhaled air can be deposited in the respiratory system [114]. When inhaled, the particles in the air come to equilibrium with body temperature and humidity, and, as a consequence, the movement of large particles becomes restricted [103]. Depending on the size of the inhaled particles, they are categorized as “extrathoracic,” which cannot pass the larynx; “thoracic,” which are particles that reach beyond the larynx (relates to particles with Dae < 10 μm); and “respirable,” which are particles that manage to pass into the pulmonary or alveolar region (relates to particles with Dae < 4 μm) [114]. Fine particle pollution can enter the body through inhalation airflow, cross the respiratory tract and reach the alveoli, where it triggers an inflammatory response that reduces the immune system’s ability to respond. Further, once in the lungs, PM can enter the bloodstream and spread to other organs.

The direction of the airflow within the respiratory system and the velocity of inhaled air impact particle deposition and diffusion. At the point of inhalation and when air comes into the tracheobronchial region, the linear velocity of the inhaled air can be relatively high. However, compression of residual gases within the alveolar sacs slows down the airflow speed, and, therefore, the velocity of the air in the alveolar region is minimal [110]. PM is composed of both soluble and insoluble particles, and it is assumed that the sizes of the particles get altered in the alveolar region due to their hygroscopic nature. Depending on the modifications to the PM, these could either be deposited in the lung or get removed with exhaled breath [115].

The United States Environmental Protection Agency (USEPA) states that only PM < 10 μm has the ability to be deposited in the trachea-bronchial and alveolar regions, and, consequently, causes the maximum danger by way of inhalation [116]. However, during exercise, breathing is mostly done by the mouth, and, as a result, larger particles (PM > 10 μm) can be deposited in the tracheobronchial airways [103]. Madureira et al. reported that 3-month old infants had 4-fold higher inhalation doses of ultra-fine particles than their mothers [112]. PM_10_ was predominantly deposited in the head region, whereas deposition of PM_2.5_ and ultra-fine particles occurred in the pulmonary area. Lower right lobes demonstrated a high susceptibility to respiratory problems because they received higher PM deposition than upper, lower left, and middle lobes. Meanwhile, the elderly is also a group susceptive to particulate matter, since they spend more time indoors. Almeida-Silva et al. used computational models to measure particle transport and deposition in the human respiratory tract of the elders [117]. The results shown that after 5 years of continuous exposure to the average particle concentration, 258 mg of all particles are deposited on the surface of the alveoli of which 79.6% are cleared, 18.8% are retained in the alveolar region, 1.5% translocate to the hilar lymph nodes, and 0.1% are transferred to the interstitium. Additionally, Segalin et al. found that respiratory deposition of PM_2.5_ was almost 25% higher in male than female elderly [118].

Whether or not bioaerosols are inhaled depends on their infectivity, airborne concentration, immunogenicity, and particle size [119]. Carpeted homes have higher concentrations of Al, As, and transition metals such as Cd and Cr than non-carpeted homes, which indicates the possibility of inhalation exposure of these elements [113].

Respiratory absorption can be summarized as follows: deposition of particulate matter (PM_10_) in the upper respiratory tract, fine particles (PM_2.5_) in the lower respiratory tract, followed by ultrafine particles (UFPs < 100 nm) in alveoli. Few scientific data exist concerning human toxicity from inhaled fungal toxins [102]. Methodologies for measuring indoor bioaerosol exposure and health risk assessment are not well standardized and, therefore, additional studies are needed on exposure-disease and dose-response relationships in humans.

### 3.2. Cutaneous Absorption

Skin, which covers the entire human body, is considered the largest organ of the body. It is made up of three main layers: epidermis, dermis, and hypodermis [120]. PM in indoor air can be deposited easily onto the skin or absorbed through the skin because it is exposed to the environment [121]. Dermal exposure also depends on dust adherence to skin, which is adapted from guidelines on dermal contact with soil and varies from 0.004 to 0.01 mg/cm^2^ for different parts of the body of children indoors and from 0.02 to 0.80 mg/cm^2^ for adults based on activity. Indoor environments are rich in semi-volatile organic compounds (SVOCs). Garrido et al. and Weschler et al. showed that the concentrations of SVOCs absorbed via direct air-to-skin dermal uptake could be comparable inhalation intake [39,122]. Williams et al. showed that perspiration-induced absorption of pesticides into the human body occurred through the dermal contact [123]. The epidermis stratum corneum is the outermost skin layer and is the primary barrier that prevents environmental pollutants from entering the body. Skin is the first defense immune system that protects the human body from toxic substances [120]. PM gets absorbed by percutaneous penetration, which causes local toxicity in the skin and systemic toxicity in other organs [124]. The four different ways that PM penetrates the skin are by mechanical delivery, an intracellular route, a transcellular route, and through the trans-follicular route [120]. Hair follicles that create pores in the skin help the PM to get into the body through the trans-follicular route [125].

### 3.3. Hand-to-Mouth Behavior

The main pathways of human exposure to PM are skin exposure to PM and particle ingestion, hand-to-mouth behavior, and food containing. However, both the rate of particle ingestion and the size of particle ingested are highly uncertain. Finer particles can stick to the hands, so both ingestion and skin contact are major problems. However, there is little consensus on how small the particulates should be to stick to your hands, since the airborne particles range in size over five orders of magnitude (from about 0.001 μm to about 100 μm) [41]. Unsurprisingly, children consumed much higher rates of particulate matter than adults because they liked to play on the floor and put their hands and non-food objects in their mouths more frequently. This behavior, combined with their smaller body size, makes exposure to chemicals through PM more important for children than for adults [126].

### 3.4. Digestive System Absorption

Absorption of PM by the digestive system can occur in two ways: directly by diet (direct consumption of food and drinks that are enriched with PM) or indirectly into the gastrointestinal tract through the expulsion of particles removed from the lungs via mucociliary transport [127,128,129,130]. It is estimated that roughly 50% of the inhaled dose could reach the intestinal tract. Therefore, researchers have pointed out the necessity of recognizing ingestion as an essential human exposure pathway to PM pollution. Research needs to be conducted on both the ingestion mechanisms and constraints to ingestion [110]. The absorption of this PM can be the epithelial lining of the small intestine, colonic epithelium, or the stomach. Researchers have identified links between the ingestion of PM and different diseased conditions in the digestive tract, but, at present, data on the effect of PM on the digestive tract are elusive [128]. More research is needed to identify how the chemical nature and sizes of the particles affect the rate of absorption of PM along the digestive tract.

Wang et al. estimated the ingestion of tetrabromobisphenol A and eight bisphenol analogs, including bisphenol A, in 12 countries [131]. The highest median estimated daily intake (EDI) of bisphenols through dust ingestion was observed in Greece, Japan, and the U.S. They showed that the EDI for infants and toddlers was high, which indicated that dust ingestion was a significant exposure pathway.

It is difficult to quantify the amount of PM a person gets into the body by staying indoors. Whether the uptake is through inhalation, dermal exposure, or absorption through the digestive tract is a personalized matter, which varies with individual human beings and the particular indoor environment (home, school, office, or another type of working place) and the length of stay in the environment. The nature of the chemical and biological constituents of the PM also needs to be considered.

## 4. Characteristics of Indoor PM

The chemical composition of indoor PM is determined by the sources of PM and chemical processes that occur both indoors and outdoors. The primary constituents of PM include inorganic metallic compounds, organic compounds of biological origin, inorganic carbonaceous material (including black carbon and elemental carbon), sulfate, nitrate, ammonium, and other ions (Table 1).

Chemical compounds in PM are grouped into various categories that consist of organic carbon (OC), elemental carbon (EC), carbonate carbon (CC), non-sea salt sulfate (NSS-SO_4_^2–^), nitrate (NO_3_^–^), ammonium (NH_4_^+^), sea salt, mineral dust, and non-dust elements [140]. Generally, the NSS-SO_4_^2^ fraction is estimated from the difference between the total sulfate and the sea-salt fraction of SO_4_^2–^. Sea-salt concentrations are estimated from soluble sodium concentrations [141]. Mineral dust components are determined by summing Al_2_O_3_, SiO_2_, CO_3_^2–^, Ca, Fe, K, Mg, and Mn. Non-dust elements include common trace elements (i.e., Cu, Ni, Pb, V, and Zn), and they are generally derived from atmospheric origin or attributed to atmospheric pollution [142]. Typically, indoor PM consists of approximately 50% organic carbon, 3% elemental carbon, 30% sulfates and nitrates, 15% ammonium ion and water, and 1% total metal content, with more than two-thirds of that being iron.

Wang et al. noticed that NSS-sulfate in PM_10_ and PM_2.5_ contributed 95% of the total sulfate in Guangzhou, China, suggesting a significant anthropogenic origin of the PM [143]. The primary sources of the sulfate in Guangzhou may be attributed to the release of SO_x_ from sulfuric-acid-manufacturing industries and the generation of sulfate compounds in coal-fired power plants and their subsequent utilization in construction and agricultural industries. Some fraction of the SO_x_ released through anthropogenic sources is oxidized to secondary aerosols of sulfate compounds [144].

The sources and composition of PM vary with PM particle size. PM_10_ consists predominantly of insoluble, Earth-crust-derived compounds such as iron and aluminum oxides; biological materials such as pollen, fungi, and bacteria; and sea salts. PM_2.5_ is derived mainly from combustion-related sources and consists of carbon, hydrocarbons, and sulfur and nitrogen [2].

In addition to PM, indoor air and ambient air also contain a range of gaseous pollutants that include carbon monoxide, sulphur dioxide, and nitrogen oxides derived from both indoor (e.g., combustion) and outdoor (e.g., bushfire) activities. Indoor PM is generally enriched with chemical additives, such as phthalate plasticizers, organophosphates, brominated flame retardants, and fluorinated surfactants used in products that are part of the indoor environment. These pollutants associated with PM cause health effects when the indoor occupants inhale PM.

The chemical elements in PM can be divided into two major groups: Earth-crust elements derived from soil (i.e., soil tracers) and introduced elements derived from human activities (i.e., anthropogenic tracers). Earth-crust elements include Na, Al, K, Mg, Ca, Fe, Ti, and Mn, which are derived primarily from geological sources, whereas V, Cr, Cd, Ni, Cu, Pb, Zn, As, Sn, and Se can be derived from both natural Earth-crust sources and anthropogenic sources [145]. These elements can be used to fingerprint the anthropogenic sources of indoor PM.

Several methods are used to fingerprint the most likely indoor PM [146,147]. For example, enrichment factors are calculated for individual elements in terms of their average concentration in the Earth’s crust. Aluminum is commonly used as a reference for these elements because of the minor contribution of Al as a potential pollutant and its major contribution to the Earth’s crust. The enrichment factor (EF) of an element *E* is defined according to Equation (1):EF = (*T*/*R*)_air_/(*T*/*R*)_crust_(1)
where *T* and *R* represent the concentrations of the tested and the reference element, respectively, if the EF approaches 1, then the Earth’s crust is considered as the dominant source of the tested element. Considering the variation in the Earth-crust composition (Table 2), EF > 5 in PM indicates that non-crustal anthropogenic sources contribute a significant portion of PM’s element.

For example, Alves et al. measured high EF values (>10) for Pb, As, Cu, and Zn in both indoor and outdoor PM, which suggested that there was a contribution of anthropogenic activities to both the outdoor and indoor environments [149]. The data also indicated that air infiltration affects the nature and composition of indoor PM. Furthermore, the mean outdoor concentrations of many of these metals for both coarse and fine PM exceed that of indoor PM, which indicates that air infiltration makes a significant contribution in lowering the enrichment of these metals in indoor PM [141].

Wang et al. noticed that Cd and Se exhibited the highest enrichment factors (>10,000) in indoor PM collected from the Guangzhou region. Pb, As, Sn, and Zn also showed high enrichment factors (>100) [143]. Ni, V, Cr, and Cu appeared to be moderately enriched (10 < EF *<* 100). The high enrichment of these elements indicates that non-crustal sources, including pollution emissions, contribute primarily to elemental loading in indoor PM. For example, the high ER for Cr reveals a range of pollution sources, including coal combustion and tannery sludge incineration [151]. While non-crustal V is derived mainly from heavy fuel oil combustion, metal smelting, and fossil fuel combustion are the likely sources of non-crustal volatile metals such as Cd, Zn, and Pb [152]. Furthermore, elements with high EF values such as Cr and V generally have low concentrations in PM. Elements originating from the Earth’s crust and sea salt, such as Na, Mg, K, Ca, Ti, Mn, and Fe, have high concentrations in PM, but low enrichment factors (<5), which indicates an insignificant contribution from anthropogenic sources of these elements in PM; airborne Earth-crust dust is the primary source of these elements in PM [150].

## 5. Health Effects of Indoor PM

### 5.1. Overall Impact

In the presence of indoor sources, PM levels can rise rapidly to several orders of magnitude greater than outdoor levels [153,154,155]. As a potential health hazard, indoor exposure to PM has received increased attention in recent years because people spend almost 90% of their time indoors [153]. Brauer et al. have found that 99% of the population in south and east Asia live in areas where the WHO Air Quality Guideline for PM_2.5_ is exceeded [156]. Exposure to indoor PM has been identified as the cause of respiratory infection, allergic symptoms, cardiovascular disease, adverse birth outcomes, and neurological and cognitive disorders [157,158,159]. Epidemiological studies have found that mortality and morbidity of respiratory diseases rose as the PM concentration increased [160,161]. Long-term exposure to PM less than 2.5 μm in diameter (PM_2.5_) is associated with chronic conditions such as cardiovascular and respiratory diseases and cerebrovascular complications, leading to reduced life expectancy [162]. Short-term exposure to PM_2.5_ can also cause a variety of health impacts including exacerbation of asthma and increases in respiratory and cardiovascular hospital admissions and mortality [162,163]. Additionally, it was also found that mortality generated by short-term PM_2.5_ exposure was influenced by season, region (urban and rural), and co-pollutants [164].

The main components of indoor PM are inhalable, which can penetrate the chest area of the respiratory system and cause adverse health effects [13]. Exposure to indoor PM will affect lung development, especially in children, including reversible deficits in lung function, chronically reduced lung growth rate, and a deficit in long-term lung function [13,165]. Moreover, some toxic pollutants, such as heavy metals and polycyclic aromatic hydrocarbons (PAHs), are also attached to the surface of indoor PM, and they pose a severe threat to human health [166,167]. PAHs adsorbed on the surface of PM have been shown to have carcinogenic and mutagenic effects [168]. Heavy metals and PAHs, individually or in concert, damage the double helix structure of DNA, leading to genetic mutations. MAC releases cytokines, such as growth factors, which alter the cell cycle and cause cells to divide forever. Thus, tumors can form. A survey has confirmed the presence of particulate polycyclic aromatic hydrocarbons (PAHs), such as bingo pyrene, in the air of Beijing. All air samples are highly mutagenic [169]. Even at low doses, heavy metals can have severe effects on neurodevelopment [170]. To control its adverse health impacts, it is essential to explore the composition of indoor PM. Studies have found that the elderly, children, and pregnant women are more susceptible to PM than others and PM concentrations vary in different indoor environments. As such, the further discussion of this issue is of great significance for removing indoor PM and reducing the exposure of susceptible groups to PM.

### 5.2. Harm of Main Components of PM to the Human Body

In general, toxicity of particulate matter refers to the absorption and distribution of chemical components of the particle, which, in addition to carcinogenicity and mutagenicity can cause adverse health effects throughout the body [171]. According to recent studies of indoor PM done at different places [172,173,174], the components of indoor PM can be roughly divided into metals (e.g., Fe, Ni, Zn, and V), inorganic compounds (e.g., sulfate, nitrate, and ammonium), and organic compounds (e.g., PAHs, volatile organic compounds (VOCs), and soot). Numerous studies have reported that some trace elements (e.g., Fe, Ni, Zn, V, Pb, As, Se, Cd, and Hg) may cause damage to cells, tissues, proteins, and DNA and change cell permeability by inducing the production of reactive oxygen species (ROS) [175,176,177]. For example, an extensive increase in ROS can be caused by Fe in vivo, causing oxidative damage and inflammation [178]. Gilli et al. found that oxidative DNA damage attributed to PM is also related to the Fe content [179]. Magnani et al. studied the effect of transition metals in air particles on pulmonary oxygen metabolism and found that Ni could cause metabolic changes [180]. Additionally, Cu, K, Mn, Zn, V, and Ni are associated with increased odds of hospital admission for cardiovascular disease, which can lead to death in severe cases [181,182,183,184].

Inorganic substances are also common pollutants in indoor air, among which nitrate, sulfate, and ammonium particles are representative. Most sulfate and nitrate in PM originate from the atmospheric oxidation of SO_2_ and NO_x_ emissions, mainly in the form of aerosols (e.g., (NH_4_)_2_SO_4_, NH_4_HSO_4_, NH_4_NO_3_, and partially neutralized salts). Ammonium ions are mainly involved in the neutralization reaction of sulfuric acid and nitric acid. Sulfate in the atmosphere can increase the deposition of toxic compounds in the lungs, affecting breathing conditions. A 1% variation in sulfate was associated with 0.117% variation in respiratory diseases [14,185]. Further, sulfates increase ROS levels, and long-term exposure to sulfates may cause oxidative stress, increasing the risk of many vascular diseases [184]. In addition to sulfates, it was found that the increase of acute cardiovascular hospitalization rate was also related to the increase of nitrate concentration [186]. Additionally, the mortality rate of the elderly was significantly associated with nitrate concentrations [187,188].

In addition to the inorganic component, the organic part of PM is formed in a complex and poorly understood way, and it is also known as organic aerosols, such as PAHs, VOCs, and soot [174,189]. PAHs, which are rich in carbon and hydrophobic, easily cross cell membranes, and quickly enter cells. Then, the PAHs from a harmful intermediate inside the cell with a ring of active epoxides, and, when a gene mutates under the circumstances of a gene polymerase error, the PAHs randomly select locations in the genome, in some cases leading to cancer [190]. Long-term exposure can affect women’s reproductive health, cause proteinuria, and even lead to lung cancer [191,192,193].

Another critical component, VOCs, are organic substances that can easily evaporate under ambient air conditions, and most of them are toxic. The VOCs can cause skin irritation, cancer, respiratory diseases, chronic obstructive pulmonary disease, bronchial asthma, and systolic plus diastolic hypertension [194,195,196]. The increased concentration of VOCs will lead to decreased respiratory and lung function in children and cardiopulmonary dysfunction in susceptible populations [197].

The levels of PM found in the indoor environment are mainly brought in by ventilated airflows or produced by burning indoors for heating and cooking. Soot is formed through a series of reactions in which small free radicals (e.g., OH, O, H, CH, and CH_2_) cause chemically induced combustion and fuel decomposition, producing larger hydrocarbon free radicals and PAHs. Some soot is cytotoxic and has adverse effects on cardiovascular and lung health. PM penetrates deep into the lungs and enters the bloodstream, causing high blood pressure and damage to blood vessels. Once in the bloodstream, PM can spread to other organs, such as the heart, damaging their cellular structure and function [198].

Compared with PM_10_, PM_2.5_ with smaller particle sizes has larger specific surface area and larger adsorption capacity, and toxic heavy metals are more likely to bind to PM_2.5_ [199], and so do acid oxides, organic pollutants and pathogenic microorganisms. Dacunto et al. found that the proportion of trace metal (transition metal) elements and carcinogenic polycyclic aromatic hydrocarbons in PM_2.5_ was almost twice that of PM_10_. At least 60% of PM_2.5_–PM_10_ is reported to be deposited on the outside of the chest [172,200].

Particles with a size ranging from 1 to 2.5 μm mainly deposited in bronchial and alveolar, and some particles remained in lung tissue for a long time, forming lung interstitial lesions. PM_0.1_ can invade alveolar and stay in it, and then quickly enter blood circulation system through breathing, and finally flow into human kidney, liver, heart, brain, and other organs. In conclusion, PM_2.5_ is more harmful to human health than PM_10_.

### 5.3. Harm to Different Groups

Because of the different occupational and physical characteristics of people, their susceptibility to PM is also different. Therefore, it is necessary to explore the different reactions of diverse populations to PM.

Most epidemiological studies have focused on the health effects of indoor PM on susceptible populations, namely pregnant women, children, or the elderly [138]. For pregnant women with long-term exposure to high levels of PM, the developing human fetus may be at risks, such as adverse birth outcomes, preterm birth, term low birth weights [159,201], congenital disabilities, stillbirths, and respiratory disease [202,203,204]. Epidemiological studies have shown that exposure to pollutants in the indoor environment is associated with respiratory diseases in children, such as wheezing, asthma, and rhinitis, and it can lead to disease in later life [205,206,207]. Mousavi et al. showed that exposure to air pollution in childhood increased the susceptibility to Alzheimer’s disease and Parkinson’s disease in adulthood [208]. Not just for children, but also for the elderly, exposure to indoor PM may be the most significant public health burden in terms of risk to health [209,210]. Chen et al. studied residential areas with long-term exposure to PM_2.5_ and PM_2.5–10_ and found that the decline of lung function in the elderly (aged 65 years or older) was related to PM [211]. Recent studies suggest that PM is associated with low bone mineral density and osteoporosis-related fractures [212]. Exposure to PM is associated with cognitive deficits, oxidative stress, neuroinflammation, and neurodegeneration [213]. Several metals, including aluminum, arsenic, cadmium, lead, manganese, and mercury, have been shown to affect the nervous system, while the general accumulation of metal ions in the brain can exacerbate oxidative stress and neuronal damage [214].

In addition to the relatively severe impact on susceptible populations, there is damage to people who, unavoidably, have long-term exposure to high PM concentrations. Indoor cooking has been considered one of the most important indoor PM [215]. Studies have shown that emissions from cooking can harm human health, leading to lung toxicity, immune-toxicity, genotoxicity, and potential carcinogenicity in the human body [216,217,218]. Particularly for individuals exposed to indoor cooking fumes, such as cooks, workers, and restaurant customers, health will be adversely affected [219]. Pan et al. suggested that kitchen staff is more likely to have oxidative stress than service area workers [220]. Bigert et al. found that female cooks, restaurant and kitchen assistants, and wait staff showed a statistically significant increase in myocardial infarction risk [221]. Welding can produce high amounts of fumes containing ultrafine particles with Mn [222]. Racette et al. observed that some might develop Parkinson’s disease 17 years earlier than the general population in welding populations [223].

Studies have shown gender differences in indoor PM exposure. Sears et al. studied people living near coal-fired power plants with coal-ash-fly-storage facilities and found that women were more susceptible than men to impaired cognitive ability and associated with PM exposure [224]. Additionally, Gregory et al. disclosed that the distribution of multiple sclerosis prevalence was related to PM_10_ concentration, especially in women [225]. In offices, female employees, and particularly those suffering from allergies, reported more sick-building-syndrome symptoms (e.g., sneezing, cough, tiredness, and irritability) than their male counterparts [226]. Contrary to the above, men are more likely than women to develop symptoms. In exploring the effects on the human brain of aerosols produced by exposure to electric frying, it was found that the aerosol response of the brain to electric frying occurred in males rather than females [227]. Weichenthal et al. analyzed the relationship between PM_2.5_ and non-accidental cardiovascular mortality and found that male cardiovascular mortality may be related to PM_2.5_ exposure, while female cardiovascular mortality had no similar association [228]. The differences in PM symptoms between men and women may be related to gender-specific behavior patterns [229].

The range of the effect of PM on health is broad, but it mainly affects the respiratory tract in children and cardiovascular function in the elderly. The impact of PM on different populations shows a need to improve health in the general population, to control the sources of PM, and to raise awareness of the potential impact of household pollutants.

## 6. Mitigation of Exposure to Indoor PM

### 6.1. Standards of Indoor PM

To reduce exposure to indoor PM and its adverse health impacts, many national organizations and influential worldwide committees (e.g., WHO) have stipulated mass standards and guidelines that coincide with desired indoor air quality. Indoor air standards to evaluate an acceptable quality of air are generally defined by various agencies, causing significant regional differences. Dai et al. installed long-term IAQ sensors in 117 homes in all climate zones of China in order to able to measure indoor PM_2.5_ and CO_2_ concentrations consecutively [230]. Using this indoor and outdoor data, local polynomial regression fitting was used to determine the relationship between indoor CO_2_ and PM_2.5_ concentrations and the corresponding outdoor parameters. Maleki et al. tested the potential of an artificial neural network (ANN) algorithm to estimate hourly air pollutant concentration parameters and two air quality indices, the air quality index (AQI) and the air quality health index (AQHI) [231]. Air quality often has negative health, socio-economic, agricultural, and political consequences. Meteorology and pollution sources are two basic factors affecting air quality. The method can be used to predict the spatial and temporal distribution of pollutants and air quality index. Main standards and guidelines formulated by global institutions are summarized in Table 3, and they guide the development of effective strategies.

In China, the previous Indoor Air Quality Standard (GBT 17095-1997) only provided a limit for PM_10_ and stipulated that the daily permissible maximum concentration of PM_10_ should be 0.15 mg/m^3^. With increased knowledge about indoor air pollution, PM with sizes <2.5 and 10 µm (PM_2.5_ and PM_10_, respectively) are now both listed as common indoor air pollutants. A standard and guidelines (JGJ/T 309-2013), issued on 2 July 2013, require that the daily average concentration of indoor PM_2.5_ should be <75 µg/m^3^. As early as 1996, Singapore formulated a control standard for PM_10_ in office air, and the value was 150 µg/m^3^ [233]. However, a control standard for indoor PM_2.5_ in Singapore is still lacking. In 2012, Canada established the Residential Indoor Air Quality Guidelines, which state that PM_2.5_ needs to be monitored, with a limit of 100 µg/m^3^ as a 1 h average (Short-Term Exposure) and 40 µg/m^3^ as an 8 h average (Long-Term Exposure). The National Ambient Air Quality Standard (NAAQS), set by USEPA, stipulates 35 µg/m^3^ and 15 µg/m^3^ for 24 h and annual periods, respectively, for exposure to PM_2.5_. For PM_10_, the US EPA [236] states that the permissible value is 150 µg/m^3^ over an average of 24 h. The United Kingdom, Finland, and Germany do not have PM_2.5_ monitoring [237]. However, these European countries follow the guidelines set by the WHO with values of 25 µg/m^3^ and 10 µg/m^3^ for 24 h and annual averages, respectively. Because various international institutions set these standards and guidelines, they are not in unity. Therefore, a comprehensive investigation should be carried out to develop uniform standards and practical strategies to mitigate exposure to indoor PM. Mass concentration predictions have a key role to play in making decisions about atmospheric resources [243]. This has had adverse health effects, such as high morbidity and mortality rates from cardiovascular and respiratory diseases. For this cause, it is crucial to avoid air pollution in advance by improving air quality protection and ensuring effective environmental monitoring. This is very critical for people’s daily safety and the government’s decision-making on air quality regulation.

### 6.2. Effective Removal Technologies of Indoor PM

Current particulate removal technologies mainly involve filtration, adsorption, electrostatic dust removal, and the negative ion and plasma (NIP) method (Table 4). They have been applied in various aspects of life to improve air quality where humans live and work.

Generally, filtration is typically installed to eliminate PM prior to other abatement technology [253]. Five principles for removal are used in filter technology, including interception, diffusion, inertia, gravity, and the electrostatic force (Figure 5) [244]. In particulate removal by filtration, bag type dust collectors, particle layer dust collectors, and carbon-based air filters are three typical, traditional methods [245,254,255]. Adsorption is often used in conjunction with filter-particulate removal in air purifiers to reach superior indoor air quality. Disadvantages include a compromised efficiency at a high relative humidity [256], the need for periodic replacement of adsorbents to prevent re-entry of waste pollutants into the atmosphere, and that airborne bacteria may thrive on carbon sorbents.

Some carbon-based materials, e.g., activated carbon fiber and granular activated carbon, with the virtues of high density, low ash content, and high absorption capacity, have been widely used for removing molecules, gases, and vapors from indoor air [248].

Electric-collection technology can be applied in filtration technology to improve the efficiency of filter-particulate removal. By applying electrostatic forces, air filters can capture PM without high-density micropores, thus leading to a sharply decreased filter thickness and higher filtration removal efficiency [257]. On the basis of these previous findings, a high-efficiency rotating TENG (R-TENG) enhanced polyimide (PI) nanofiber air filter was developed for PM removal in ambient atmospheres by Gu et al. [251]. (TENG stands for triboelectric nanogenerator.) Charged positively by the R-TENG, an electric field was formed around a stainless-steel mesh and PI-nanofiber, thus significantly improving the PM-capturing efficiency of the filter due to electrostatic filtration [258]. However, electronic filters can generate hazardous charged particles, an obstacle for the extensive application of electrostatic-particulate-removal technology [259].

In addition to hazardous gases, indoor particles also can carry large amounts of harmful viruses and bacteria. Therefore, to ensure indoor human health and reduce noise pollution from filtration, new particulate-removal technologies that can provide such problems are imperative. Negative ion and plasma (NIP) methods are being developed due to their excellent ultra-fine particle removal and sterilization without making noise. The principles of the NIP methods are similar. Both are charged by an external force to neutralize the particles and coalesce them to form larger particles that settle [260]. Nowadays, the main application of the NIP method is plasma dust collection [252]. However, the PM is not removed but only adheres to the nearby surface, and it is easy to raise dust again. Therefore, the widespread application of the negative ion and plasma method is restricted.

### 6.3. Strategies to Reduce Exposure to Indoor PM

Rational and effective methods need to be taken to mitigate exposure to indoor PM for the sake of human health and a comfortable indoor environment. Based on the characteristics and sources of indoor PM, control strategies can be targeted at two sources: ambient PM and domestic PM.

#### 6.3.1. Control Strategies for Ambient PM

(1)Building ventilation

Ventilation of households is crucial to bring indoor PM concentrations to levels similar to those of outdoors [261]. Specific measures should be considered, based on the ventilation types of buildings, which are identified as being natural, mechanical, or a mixed type of ventilation. Buildings with mechanical ventilation systems (air conditioners) show a minimum concentration of particles, which indicates that adequate household ventilation can improve indoor air quality. Building ventilation (interventions) is beneficial to mitigate indoor PM exposure, but they always incur a high energy expenditure cost. Therefore, an optimized building intervention model is needed to establish maximum efficiency and minimum energy cost [262]. When outdoor PM_2.5_ concentration is high, doors and windows should be closed. The key of the method is real-time monitoring of indoor and outdoor PM_2.5_. Also important is improving air distribution, increasing ventilation rate, fresh air (filtration) volume, regularly cleaning and disinfecting air-conditioning system, etc. In buildings with centralized or semi-centralized air-conditioning systems, the use of fresh air systems can reduce indoor PM_2.5_. Some experts believe that for public buildings, air purification facilities could be equipped in existing fresh air units, return air inlet, and blast pipes, and air cleaners also could be used directly there [261].

(2)Climate and season

In some regions, climatic and seasonal characteristics are distinct, such as in the east and west of the USA and southeast China. The variations can be used to mitigate exposure to indoor PM [263,264,265]. In China in the summer, the prevailing wind direction from the south will bring clean air from the ocean. As such, at this time, citizens should open windows and doors frequently to promote clean air exchange. In winter, windows and doors should be closed as far as possible. Industrial emissions must be reduced. The coal should be desulfurized before burning. Boilers of 30 t/h should be phased out as soon as possible. In North China, central heating should be promoted.

(3)Traffic and industries

With rapid urbanization and industrialization, particles generated by traffic and industries have aggravated the indoor air environment. To handle air pollution in heavily trafficked areas, enough urban greenspace should be guaranteed to reduce indoor levels of PM, especially PM_2.5_ [266]. In addition, vehicle emission must be controlled, and emission standards must be strictly followed. Additionally, vehicles that use new-energy technology ought to be promoted. Urban greenery and other protective procedures should be enhanced where traffic is heavy. Finally, the upgrading of petroleum refining should be accelerated to prepare for the improvement of motor gasoline and diesel.

Industrial and agricultural emissions are the two primary sources of particles that need to be reduced. Increased use of clean fuels and effective dust extraction is crucial so that industries and agriculture can reduce the generation and emission of PM. Removal of industrial dust is an issue that people are increasingly concerned about because of its health effects on humans, especially workers in factories. Nanofibrous filters, produced by electrospinning, have increasingly been used in air filtration products due to their high surface area and micro-porosity, which improve the entrapment of PM [267]. Liu et al. synthesized a transparent air filter by electrospinning, which achieved high ventilation and PM_2.5_ filtration efficiency (>95.0%) [268].

#### 6.3.2. Control Strategies for Indoor PM

(1)Smoking

The first strategy is to control smoking. The government and relevant departments should actively promote the smoking ban and enhance the public’s health awareness. In public places, such as train stations and restaurants, there should be smoking and non-smoking areas and a reasonable division of functional areas to avoid cross-contamination. One study showed that Ontario’s smoking ban saved five to seven non-smokers working in bars $5–6.8 million a year [269]. With the development of e-cigarettes, more teens are jumping on the bandwagon and using them; even though levels of some potentially harmful ingredients from e-cigarettes are significantly lower than combustible cigarette, this does not mean that e-cigarette aerosols are ‘‘harmless vapour” as industry has claimed in the past [270]. The effects of SHA vaping exposure are largely unknown. E-cigarettes are not emission-free and their pollutants can reduce indoor air quality because the air exhaled by e-cigarette smokers contains dangerous chemicals. There is a potential health concern of SHA exposure via both respiration and dermal absorption. In particular, ultrafine particles formed from supersaturated 1,2-propanediol vapour can be deposited in the lung, while atomized nicotine appears to increase the release of inflammatory signaling molecule NO after inhalation.

Overall, the increased use of e-cigarettes is worrisome and restrictions on tobacco marketing should be implemented to better protect the health of the general public. Future research should specifically focus on the long-term adverse effects of e-cigarettes on the cardiovascular system or respiratory diseases and cancer, as there is still a lack of strong evidence. There is no doubt, however, that quitting smoking is and will continue to be the most powerful way to prevent the cardiovascular and respiratory diseases caused by smoking and to protect people’s health [271].

(2)Cooking

Natural gas cooking has been identified as one of the predominant sources of indoor PM in developed countries [51,215,272,273,274,275,276]. Therefore, the development of methods to control PM generated from cooking is imperative. Kitchens must be ventilated to promote air exchange during cooking. Traditional solid fuels, such as straw, coal, and asphalt, must be replaced by clean energy as much as possible. Natural gas, methane, and electricity are good clean-energy options. Stoves must be maintained in functional order, especially in rural areas [277]. Cooking habits should be improved, and exhaust hoods are crucial for control of indoor PM during cooking [261]. Different cooking methods and ingredients, such as preferring safflower oil to olive oil. Adding salt and pepper early in cooking can also reduce the emission of PM.

(3)Indoor activities

Apart from cooking, indoor activities should be carried out to reduce PM. People should clean households regularly to remove the origins of indoor PM. Regular cleaning of the house, reasonable choices of decoration, as far as possible not keeping pets, and burning incense less, are considered good habits. Branco et al. demonstrated that improving the ventilation rate with convenient cleaning decreased the indoor PM concentration in a nursery from 79 ± 22 µg/m^3^ to 64 ± 15 µg/m^3^, which showed that regular indoor cleaning could control indoor PM [278].

(4)Indoor layout

Interior decoration, such as house plants, can improve indoor air quality and reduce indoor PM [266]. Previous studies have confirmed that the impact of woodland with rough surface on PM_2.5_ is much greater than that of grassland, while grassland has historically had a smaller effect on PM_2.5_ reduction [191,279]. However, tree-grass configurations contributed much more than tree only configurations for horizontal PM_2.5_ reduction, indicating that grassland is playing a contributory role in reducing atmospheric PM_2.5_. Vegetation has the irreplaceable role of adsorbing dust and removing harmful substances in the air. Leaf structure can be chosen for the deposition of dust [280,281]. Trees remove gas pollution mainly through leaf stomata, although some of the gas is removed from the plant’s surface. Once inside the leaf, the gas diffuses into the intercellular space and may be absorbed by the water membrane to form acids or react with the inner surface of the leaf.

Trees also eliminate pollution by intercepting airborne particles. Some particles can be absorbed by the tree, but most of the trapped particles remain on the surface of the plant. The intercepted particle is often suspended back to the atmosphere, washed away by rain, or dropped to the ground with leaf and twig fall. A widely accepted view is that trees with large leaf area density and the turbulent air movement caused by their structure can capture particulate matter. Leaves provide surfaces for removing pollutants through wet and dry deposition, adsorption, and absorption. Local decreases of temperatures may modify the rate of chemical reactions, leading to decreased ozone concentrations [282]. Green plant species differ in their ability to purify particulate air pollution. Appropriate indoor green plants for air purification are orchids, caladium, and red back laurel. Leaf hairs can trap and absorb floating particles and smoke in the air. Active green walls use ornamental plants growing along the vertical plane, coupled with mechanical air induction, to actively attract polluted air through the plant growth substrate and leaves. Additionally, with improved living standards, urban residents might buy air cleansers to reduce indoor PM levels and improve indoor air quality. HEPA air purifiers are effective at reducing the concentration of indoor air particles [283]. HEPA filter-type small air purifiers demonstrate relatively high particle removal performance, based on the high single-pass collection efficiency of the HEPA filters (>99.97% for 0.3 μm particles).

## 7. Conclusions

PM includes a mixture of solid and liquid particles suspended in air, and these particles can vary in size, shape, and chemical composition. Particles that are less than 10 μm in diameter (PM_10_ and PM_2.5_) are of major concern, because they are inhalable, thereby impacting the heart and lungs, and are associated with respiratory diseases, including asthma, chronic bronchitis, and acute bronchitis. Indoor PM is more harmful to special populations, such as the elderly, children, and pregnant women. Indoor PM originates mainly from combustion activities and the regular wearing of household furniture. It also is derived from outdoor sources, including dust particles. PM can be enriched with inorganic and organic contaminants, including toxic heavy metals and carcinogenic, volatile organic compounds. To reduce exposure to indoor PM and its adverse health impacts, the government and industry should formulate detailed and uniform indoor PM control standards based on abundant research. From a practical point of view, indoor air-purifier technologies involving electrostatic precipitation and filtration, as well as natural ventilation, are most commonly adopted to control concentrations of indoor PM. Additionally, regular indoor cleaning and suitable interior decoration, such as house plants and air purifiers, also significantly influence indoor air quality and reduce indoor PM.

Given the current knowledge about the sources, distribution, pathways, characteristics, and health effects of indoor PM, the following research areas should be pursued:Development and adoption of advanced technologies, such as the tapered element oscillating microbalance (TEOM), X-ray fluorescence (XRF), and inductively coupled plasma mass spectrometry (ICP-MS), to quantify and fingerprint sources of indoor PM.Characterization and monitoring of bioaccessibility of inorganic and organic contaminants in indoor PM.Studies on mucosal interactions of indoor PM and associated contaminants concerning their toxicity.Development and evaluation of advanced PM removal technologies involving electrostatic precipitation to mitigate the health impacts of indoor PM.As soon as possible, the government and industry should formulate detailed and uniform indoor PM control standards, based on many investigations.

## Figures and Tables

**Figure 1 ijerph-18-11055-f001:**
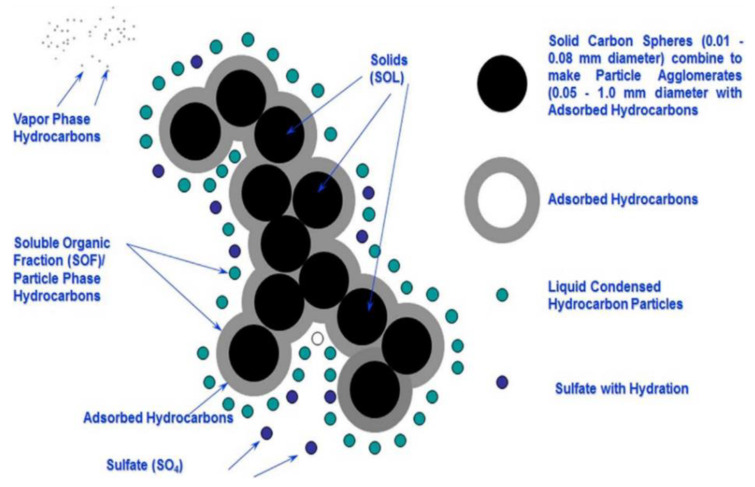
Schematic representation of PM (Source reference: [4]).

**Figure 2 ijerph-18-11055-f002:**
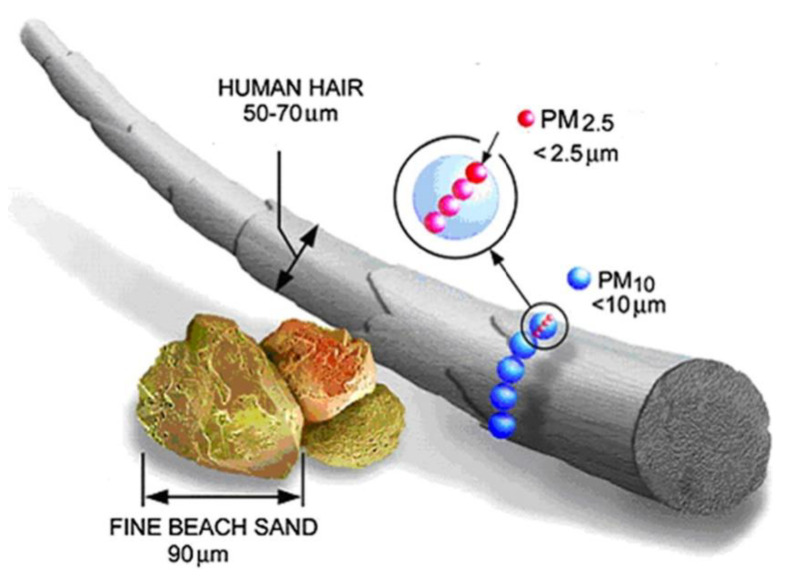
Size comparison of PM_2.5_ and PM_10_ against the average diameter of a human hair (~70 μm) and fine beach sand (~90 μm) (Source reference: [1]).

**Figure 3 ijerph-18-11055-f003:**
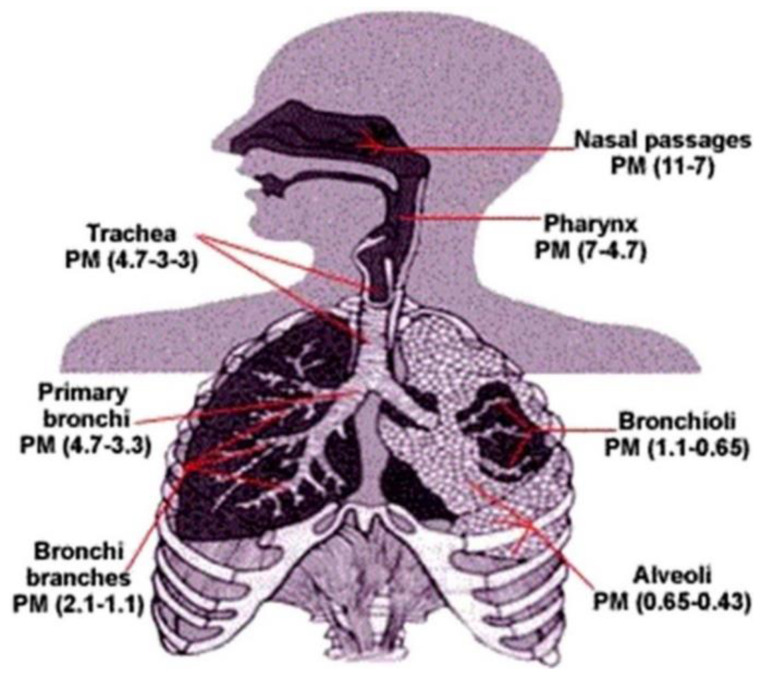
Deposition potential for particles of varying sizes (Source reference: [1]).

**Figure 4 ijerph-18-11055-f004:**
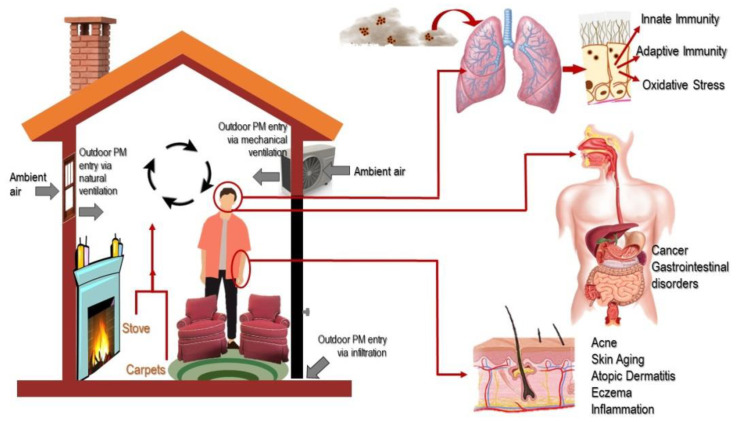
The main pathways of exposure to indoor PM.

**Figure 5 ijerph-18-11055-f005:**
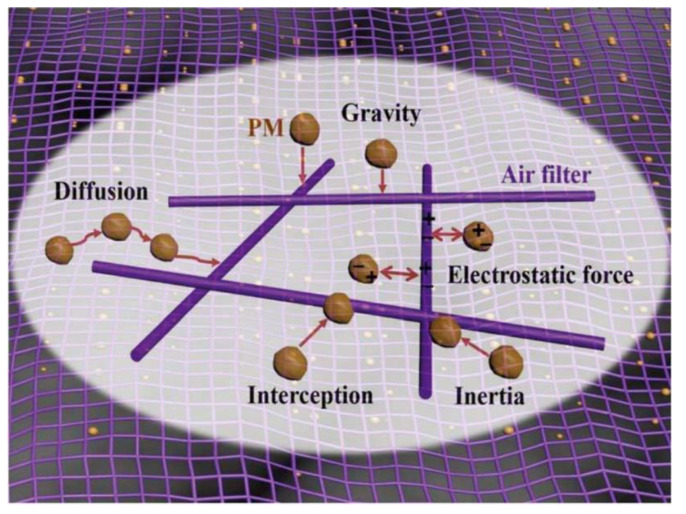
The five main capturing mechanisms for various kinds of PM (Source reference: [244]).

**Table 1 ijerph-18-11055-t001:** Selected references on the composition of indoor PM.

Region	Level (μg/m^3^)	Composition							References
		Carbon (μg/m^3^)	Non-Sea Salt Sulfate (NSS-SO_4_^2–^) (μg/m^3^)	Nitrate (NO_3_^–^) (μg/m^3^)	Ammonium (NH_4_^+^) (μg/m^3^)	Sea-Salt (μg/m^3^)	Mineral Dust (μg/m^3^)	Non-Dust Elements (μg/m^3^)	
A classroom in Xi’an, Northwestern China	VFPs = 35.4	11.94	3.60	0.94	0.20	0.52	0.38	0.15	[132]
Elementary schools in Curitiba, Brazil							7.41	0.18	[133]
Biology Department Building, University Kebangsaan Malaysia	PM_10_ = 271							3224.84	[134]
Indoor go-kart facilities	PM_10_ = 4.9 − 34.9 PM_2.5_ = 2.3 − 29.2						2.10	0.16	[135]
Nine offices in the province of Antwerp, Belgium	PM_2.5_ = 0.09		2.33	0.82	0.76		0.73	0.073	[136]
Haidian district is close to the fourth ring road of Beijing	PM_2.5_		20.54	27.51	18.73	0.43	4.78		[137]
Broechem is a village (12 km^2^) located in the province of Antwerp	PM_2.5_ = 24.8 PM_1_ = 15.7		36.4	45.7	22.1	4.7			[138]
A peri-urban area about 40 km from Rome	PM_2.5_ = 16.7 PM_10_ = 27.6	12.72	4	1.32	0.61	0.88	4.4	230.56	[139]
Located in the NE of the Iberian Peninsula	PM_2.5_ = 37	11.3	1.4	0.72	0.48	0.34	9.76	0.075	[8]

Note: Carbon include organic, elemental and carbonate carbon. The non-sea salt sulfate is calculated from the measured sulfate minus the sea-salt fraction of SO_4_^2–^. Sea-salt concentrations are generally calculated from soluble sodium concentrations. Mineral dust is considered as the sum of Al_2_O_3_, SiO_2_, CO_3_^2–^, Ca, Fe, K, Mg and Mn. Non-dust elements correspond to the sum of the common measured trace elements (i.e., Cu, Ni, Pb, V, Zn) other than geological ones.

**Table 2 ijerph-18-11055-t002:** Selected references on the enrichment ratio of various elements in indoor PM.

Region	EF		Source	References
	Earth Crust/Soil	Non-Earth Crust		
Universiti Kebangsaan Malaysia, Building 1	<1	2–5	Undefined sources, Crustal sources, Indoor-induced sources, urban origin sources and Earth’s crust	[148]
Universiti Kebangsaan Malaysia, Building 2	<1	2–5	Undefined sources, Combustion sources, biogenic sources, anthropogenic sources, crustal source	[148]
León (Spain) university cafeteria	<5	>10	Building materials, consumer products, and human activities	[149]
Broechem is a village located in the province of Antwerp, Belgium	0.1–2.4	>100	Traffic and domestic heating, the harbour of Antwerp, a large petrochemical plant, a municipal waste incinerator, and a nonferrous plant to the south of Antwerp	[138]
Nine offices in the province of Antwerp, Belgium	<10	10–1000	Outdoor influences, indoor respirable suspended particulates	[136]
Six schools located in Chañaral, Chile	5–20	<2	Industrial and mining activities	[136]
Residential and commercial buildings of Doha city, state of Qatar	1.04–3.03	1.94–63	Outdoor mineral particles, non-exhaust traffic emission, industrial sources, the influence of indoor activity such as smoking.	[135]
Guangzhou city, China	<5	10–1000	Coal combustion and sewage sludge incineration	[133]
Xian city, China	<5	10–30	Building construction, paved road dust, fresh soil dust	[150]

Earth crust elements or soil tracers and anthropogenic tracers; Earth crust elements: Na, Al, K, Mg, Ca, Fe, Ti and Mn; Non-Earth crust or anthropogenic: V, Cr, Cd, Ni, Cu, Pb, Zn, As, Sn, and Se.

**Table 3 ijerph-18-11055-t003:** Standards and guidelines for PM_2.5_ and PM_10_.

Country	Value	Organization	Reference
China	0.15 mg/m^3^ of PM_10_; 75 µg/m^3^ of PM_2.5_	AQSIQ and CABR	[232] and (JGJ/T 309-2013)
Singapore	150 µg/m^3^(in office) ^1a^ of PM_10_	Institute of Environmental Epidemiology	[233]
Australia	N/A of PM_2.5_ 90 µg/m^3^ of PM_10_	N/A The National Health and Medical Research Council	[234]
Canada	100 µg/m^3^ as 1 h average (Short-Term Exposure) 40 µg/m^3^ as 8 h average (Long-Term Exposure)	Health Canada	[235,236,237,238]
US	3 mg/m^3^ as 8 h average (Ceiling Level) ^1b^ 35 µg/m^3^ as 24 h average of PM_2.5_ 15 µg/m^3^ as 1 y average of PM_2.5_ 150 µg/m^3^ as 24 h average (Exposure) ^2a^	American Conference of Governmental Industrial Hygienist, 2005. NAAQS/EPA ASHRAE	[236,237,239]
Finland	<20 µg/m^3^ as 8 h average of PM_10_ 4 mg/m^3^ as 8 h average of PM_10_	FiSIAQ	[237]
Germany	50 µg/m^3^ as 24 h average of PM_10_	FiSIAQ	[236,240]
Worldwide	25 µg/m^3^ as 24 h average of PM_2.5_ 10 µg/m^3^ as 1 y average of PM_2.5_ 20 µg/m^3^ as 1 y average of PM_10_	WHO	[241,242]

^1a^ Guidelines for good IAQ in office premises (Singapore); ^1b^ Ceiling Level: Highest possible allowed value for exposure (US, ACGIH); ^2a^ Exposure: It means a continual and repetitive contact with the substance over a set period (US, ASHRAE).

**Table 4 ijerph-18-11055-t004:** Application of Removal Technologies.

Theory	Application	References
Filtration	Bag type dust collector Ultralow penetration air filters Pulse-jet cleaning of bag filters Triboelectric air filter	[244] [245] [246] [247]
Adsorption	Carbon-based materials	[248]
Electrostatic	Wet electrostatic precipitators Tube electrostatic precipitator (R-TENG)-enhanced PI-nanofiber air filter	[249] [250] [251]
NIP Technology	Plasma dust collector	[252]

## Data Availability

Not applicable.

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
