# Peer review of "Indoor Particulate Matter in Urban Households: Sources, Pathways, Characteristics, Health Effects, and Exposure Mitigation"

_ijerph, 2021, doi:10.3390/ijerph182111055_

Round 1

Reviewer 1 Report

This review article is well compiled information which had already been available in the literature and I found that this review article to similar to doi: 10.3390/ijerph17082927

In Figure 1 they need to incorporate metals too along with adsorbed carbon and sulphate and nitrates

Authors need to incorporate the recent findings of Epidemiological studies in to the review  

Author Response

Thank Reviewer #1 for his/her thoughtful comments. We have added more detailed information and made some deep discussion. Briefly, we have added some studies from other countries or regions. In addition, we have reorganized the health effects section. Especially, we have also added and explained the human activities causing indoor PM. Finally, the recent findings of Epidemiological studies were incorporated into the manuscript. We hope that you can satisfy this current version of the manuscript. The supplementary contents marked in the red color in the manuscript.

Reviewer 2 Report

This is an overall interesting and well written review of indoor particulate matter in urban households that contains an extensive and up-to-date literature search.
However, there are some suggestions that could be considered by the authors to ensure that the contents of their scientific contribution can be appreciated throughout the scientific world.
In this sense, I could not fail to notice that the Authors have preferentially used literature sources from geographic areas closest to them, neglecting many scientific contributions from other parts of the world. Furthermore, the sections dedicated to health effects are overall less detailed than those concerning environmental aspects.

I suggest to the Authors to balance the sections taking into account the two aspects that I have pointed out.

Author Response

Thank Reviewer #2 for his/her thoughtful comments and all of them are very important. Based on his/her comments, we have added some studies from other countries or regions. In addition, we ave reorganized the health effects section, and hope you will satisfy with the current version. The supplementary contents marked in the red color in the manuscript.

Reviewer 3 Report

I would like to thank authors the your effort done to compile and summarize all the information presenteed and reffered in this paper. It can be very useful mainly for many researcher. 

This review discusses the sources, pathways, characteristics and effects on  human health of indoors pollution (PM) considering also mitigation strategies. The topic is very relevant because there are few studies focused on indoors pollution and particulate matter impacts on health. The study responds to an existing gap in the published literature where a indoors PM was not considered important except in industrial activities. This paper outline and describe in a very well organized way how relevant indoors PM pollution (below PM 2.5) can be for human beings. In my opinión, it is a very good compilation of recent studies and  can be usefull for the scientific community working on these topics and for early carree scientist who can find here an excellent review to start working in this topics. The paper is well written and easy to read even if you are not an expert in the topic or english is not your native language , what makes the article even more valuable if we consider it is a review paper. Conclusion are based on the extensive review of scientific papers refferred. As I mentioned in the on line form , this article may also be considered for publication as a book chapter if the chief editor estimated that it does not fit in the journal as it is.

Author Response

Thank Reviewer #3 for his/her thoughtful comments. We have also revised the manuscript thoroughly to make it better.

Reviewer 4 Report

Many sentences in the article are not properly supported by other authors/studies, which should be improved (lines 115, 118-123, 136-139, 353-360, 360-385, 388-390, 763-765 ).

line 81 - add other studies that show this reality in several countries

lines 118-123 - state that "Cooking indoors is a well-researched source of PM" is well studied, but has only 3 references, two of which are not very recent and the other is very specific to a type of kitchen equipment that it is not used much in some cultures.

In chapter 2.2 they refer to the "Distribution characteristics of PM" but they do not describe them all, namely the biological ones.

lines 231-234 - refer to "...indoor and outdoor environmental conditions influence indoor PM levels.", but in the article only some of these conditions present some information to make this relation. They must detail and justify with other authors/studies.

3.1 - use more studies, with other populations, namely the elderly who are also a more vulnerable group. https://core.ac.uk/download/pdf/161376209.pdf

line 534 - select most recent study or do not mention that it is recent (since it is dated 2005)

line 591, 634 - correct to PM10

6.3 - they only mentioned the strategies of one country, being a systemati review they should have other examples.

format text from line 849

Author Response

Many sentences in the article are not properly supported by other authors/studies, which should be improved (lines 115, 118-123, 136-139, 353-360, 360-385, 388-390, 763-765 ).

Response: Thank Reviewer for his/her thoughtful comments and all of them are very important. Based on his/her comments, we have modified the related sentences and added some new references (See lines 124-126, lines 127-131, lines 141-151, lines 397-398, lines 410-413, lines 420-427, lines 806-808).

line 81 - add other studies that show this reality in several countries

Response: Thank Reviewer #4 for his/her thoughtful comments. We added the relevant information (See lines 81-89).

lines 118-123 - state that "Cooking indoors is a well-researched source of PM" is well studied, but has only 3 references, two of which are not very recent and the other is very specific to a type of kitchen equipment that it is not used much in some cultures.

Response: Thank Reviewer #4 for his/her thoughtful comments. To make this part clear, this section has been rewritten. In addition, the old three references have been replaced by two relatively new references (See lines 146-151).

In chapter 2.2 they refer to the "Distribution characteristics of PM" but they do not describe them all, namely the biological ones.

Response: Thank Reviewer #4 for his/her thoughtful comments. We have added the biological distribution characteristics of PM, and hope you will satisfy with the current version (See lines 240-254).

lines 231-234 - refer to "...indoor and outdoor environmental conditions influence indoor PM levels.", but in the article only some of these conditions present some information to make this relation. They must detail and justify with other authors/studies.

Response: Thank Reviewer #4 for his/her thoughtful comments. We have added the discussion in this section and introduced some new literature (See lines 260-269).

3.1 - use more studies, with other populations, namely the elderly who are also a more vulnerable group. https://core.ac.uk/download/pdf/161376209.pdf

Response: Thank Reviewer #4 for his/her thoughtful comments. We added a discussion on the distribution of indoor particulate matter in the elderly population (See lines 376-384).

line 534 - select most recent study or do not mention that it is recent (since it is dated 2005)

Response: Thank Reviewer #4 for his/her thoughtful comments. We have added some relatively new literature (See lines 577-581).

line 591, 634 - correct to PM10

Revised, as suggested.

6.3 - they only mentioned the strategies of one country, being a systematic review they should have other examples.

Response: Thank Reviewer #4 for his/her thoughtful comments. In fact, the PM control strategies listed in the section 6.3 are taken from multiple countries, not just China. To make it clear, we have removed some inappropriate expressions (See lines 787-790).

format text from line 849

Revised, as suggested.

Round 2

Reviewer 1 Report

This article can be accepted